# Protein-Gene Orthology in *Baculoviridae*: An Exhaustive Analysis to Redefine the Ancestrally Common Coding Sequences

**DOI:** 10.3390/v15051091

**Published:** 2023-04-29

**Authors:** Carolina Susana Cerrudo, Lucas Federico Motta, Franco Uriel Cuccovia Warlet, Fernando Maku Lassalle, Jorge Alejandro Simonin, Mariano Nicolás Belaich

**Affiliations:** Laboratorio de Ingeniería Genética y Biología Celular y Molecular—Área Virosis de Insectos (LIGBCM-AVI), Instituto de Microbiología Básica y Aplicada, Departamento de Ciencia y Tecnología, Universidad Nacional de Quilmes, Roque Sáenz Peña 352, Bernal B1876BXD, Buenos Aires, Argentina

**Keywords:** baculovirus, protein-genes, orthology, major occlusion body protein

## Abstract

Baculoviruses are entomopathogens that carry large, double-stranded circular DNA genomes and infect insect larvae of Lepidoptera, Hymenoptera and Diptera, with applications in the biological control of agricultural pests, in the production of recombinant proteins and as viral vectors for various purposes in mammals. These viruses have a variable genetic composition that differs between species, with some sequences shared by all known members, and others that are lineage-specific or unique to isolates. Based on the analysis of nearly 300 sequenced genomes, a thorough bioinformatic investigation was conducted on all the baculoviral protein coding sequences, characterizing their orthology and phylogeny. This analysis confirmed the 38 protein coding sequences currently considered as core genes, while also identifying novel coding sequences as candidates to join this set. Accordingly, homology was found among all the major occlusion body proteins, thus proposing that the polyhedrin, granulin and CUN085 genes be considered as the 39th core gene of *Baculoviridae*.

## 1. Introduction

Baculoviruses are a well-studied group of viral entomopathogens due to their versatile biotechnological applications in various scientific and technological fields, including agriculture and other industries such as human and animal pharma [1,2,3]. In terms of the arthropod virome, *Baculoviridae* is a viral family classified under group I of the Baltimore classification (class: *Naldaviricetes*, order: *Lefavirales*) where the nucleic acid is a single covalently closed circular and double-stranded DNA (cccdsDNA), between approximately 80 and 180 kbp depending on the isolate [4,5]. This genomic molecule is packaged in rod-shaped polar nucleocapsids (hence its name), which in turn can be enveloped in two types of lipid membranes that contain different proteins with important functions in the transduction processes in host cells. This results in two virion variants in some baculoviruses: budded virions (BVs, detected in lepidopteran and dipteran baculoviruses) and occlusion-derived virions (ODVs, detected in all baculoviruses). The ODVs are further enclosed in protein crystals that conform granular or polyhedral structures (granuloviruses -GV- or nucleopolyhedroviruses -NPV-, respectively) recognized as occlusion bodies (OBs) [6]. OBs may contain one (as reported in the GVs) or more ODVs (as reported in NPVs), and these, in turn, may contain one or more nucleocapsids depending on the baculoviral species (as occurs in some NPVs) [4].

During the natural infection cycle, larvae of susceptible Lepidoptera, Hymenoptera and Diptera consume OBs, which release ODVs in the midgut, causing primary infection. In this stage, the per os infectivity (PIF) complex located in the ODV membrane plays a crucial role in determining host range [7]. Subsequently, most baculoviruses in the infected cells will produce a generation of BVs which will initiate the secondary infection and will spread the pathogen to other tissues of the larva. The fusogenic proteins present in the BV envelopes, such as F protein and GP64, facilitate the entry of nucleocapsids into the insect cells [8]. This characteristic enables the multiplication of some baculoviruses in cell lines in in vitro cultures and support their applications in mammals [2,3]. In the final stage of the infection, a progeny of OBs will be accumulated and then, released into the environment when the affected larva dies and loses structural integrity (often due to auxiliary protein factors encoded by the virus itself) [4].

So far, hundreds of baculoviral isolates were described and nearly 300 genomes were reported in GenBank [9]. This significant diversity led to the establishment of taxonomic criteria for better study and consideration. The current classification is based on protein-gene content, phylogenetic relationships, virion morphology and host susceptibility, among others. The classification divides baculoviruses into four genera: *Alphabaculovirus* (lepidopteran NPVs), *Betabaculovirus* (lepidopteran GVs), *Deltabaculovirus* (dipteran NPVs), and *Gammabaculovirus* (hymenopteran NPVs) [5,10]. The first two sets of viruses are those with the largest number of known isolates, which allowed for the discovery and characterization of intra-group diversity and the suggestion of new taxa, such as groups 1 and 2 in alphabaculoviruses [4]. In contrast, three sequenced genomes were reported as gammabaculoviruses [11,12] and only one as deltabaculovirus [13], limiting a further analysis of the existing diversity in these genera. The unequal distribution of the known baculoviruses in each of the established genera is probably due to a greater search effort for members of *Alpha*- and *Betabaculovirus* since their natural hosts are usually major agricultural pests worldwide. 

Baculovirus genomes typically contain between 100 and 200 protein-coding genes [4,5,14], and a smaller and not yet fully described number of RNA genes that would produce microRNA-like regulatory molecules [15]. Nowadays, the protein-gene content is the target used for the classification criteria in different baculoviral species [10,16]. This led to bioinformatic studies to identify the amount of coding sequences that are shared by all members with sequenced genomes. The number of homologous genes conserved in all genomes (known as core genes) fluctuated in the past, but stabilized at 38 [17,18]. However, with the increase in new genomic data, there is a need for reanalysis, not only to reaffirm or modify the number of core genes, but also to describe which genes are essential to each of the genera and other considered taxa. In fact, in some baculoviral species, the existence of differential genes among isolates was even reported, which may have a significant impact on their bioinsecticide power when they are used as biological control agents [19]. This sequence reanalysis will make it possible to redirect and redefine functional genomics studies to better understand the infection cycle, and to add knowledge in baculoviral gene ontology that includes the activity of each protein, its location, and the biological processes in which it participates. Through this process, baculoviral proteins could be better classified in their roles such as transcription, replication, transport, gene regulation, responses to the host, assembly of BVs and ODVs or other important auxiliary functions for the generation of progeny. Moreover, reanalysis of the baculovirus protein-gene content may lead to rational improvements on the natural baculoviruses for better biotechnological use, given the various technologies available to edit their genomes [3,20] and the various fields where these viruses can be applied.

The term “gene homology” or “gene orthology” refers to the common ancestral origin of sequences from different biological entities, and there are various approaches to inferring it [21,22]. During the course of evolution and speciation processes, nucleotide changes can accumulate leading to a loss of similarity and potentially hindering the recognition of common ancestry. However, despite such changes, genes may continue to play similar biological roles. Regarding this, an exhaustive comparative genomic study focusing on the baculoviral protein-gene content with an orthology criteria is presented in this work, with the purpose of identifying the total number of protein genes that exist in the known baculoviral metagenome and their distribution in the current taxonomical groups.

## 2. Materials and Methods

### 2.1. Baculovirus Genome Data Set

Baculovirus genomes were retrieved from GenBank [9] (www.ncbi.nlm.nih.gov; last accesed on 1 June 2022) along with their annotated protein genes. To avoid unnecessarily redundant in sequence data, the most up-to-date version file was taken as inclusion criteria, corresponding to viral isolates from different regions or sequenced by different researchers. For sequences from the same viral species, geographical place and submitted by the same group of scientists, only those presenting differences in nucleotide similarity were included. Out of the 297 genomes thus selected, 1 sequence (Lymantria dispar multiple nucleopolyhedrovirus isolate T3; accession number MF311096.1) was discarded because, according to the database file, it did not encode LEF-5 (core gene) due to mutations (“nonfunctional *lef-5* due to mutation”), without an associated scientific publication endorsing this particularity. In addition, a novel sequence (SporGV) [23], obtained by our working group and not yet submitted to GenBank, was included. Therefore, a database (hereafter identified by the acronym BPDB for *Baculovirus Proteome DataBase*) containing 297 complete baculoviral genomes and proteomes was generated as an input to determine homologous gene groups and phylogenetic inferences (Appendix A). All graphs derived from the analysis of BPDB were created in R [24].

### 2.2. Ortholog Bioinformatic Pipeline for Baculovirus Genes

To infer the ortholog groups for baculoviral protein genes in BPDB an ad hoc automatized computational pipeline was generated (hereafter identified by the acronym OBP for *Ortholog Bioinformatic Pipeline*) based on Phyton scripts [25] and different bioinformatics packages (Figure 1). Briefly, OBP consists of eight concatenated instances (each of them with the return of partial results) covering close and remote homology and considers hypothetical proteins not annotated in GenBank files. The current 38 baculoviral core proteins [17,18] were used as test and fit set for OBP algorithm. The first step involved using proteinOrtho (https://www.bioinf.uni-leipzig.de/Software/proteinortho/, last accessed on 30 November 2022) [26] with BlastP, “e-value = 10^−7^” and “Minimal algebraic connectivity = 0.3” as parameters. The resulting sets of orthologs (*partial result 1* -PR1-) were then used for multiple sequence alignment (MSA) with MUSCLE 5 [27] to generate hidden Markov models (HMM) by HMM3 [28]. These HMMs were used as probes to search in BPDB, removing any proteins not found in the PR1 ortholog clusters (e-value 0.01). This process was repeated three times (without the sequence elimination step but if any initial sequence was not found, fasta file was removed) until a new set of orthologs was generated (*partial result 2* -PR2-). At this stage, new groups of orthologs were added by combining those that had at least 1 protein in common and with the requirement that there be no more than 1 protein per baculoviral proteome (without removing the groups prior to their merge). Once this was completed, the MSA was repeated using MUSCLE to generate new HMMs, and with these, the BPDB was searched again to generate a new set of results (*partial result 3* -PR3-). From here, the construction of the HMMs was repeated for each group of homologues (default e-value, 0.001) and the subsequent search in BPDB iterating until the groups of orthologs contained 297 proteins (one from each genome of BPDB), or the number of sequences contained does not change, or it loses sequences previously contained. This generated a new set of results (*partial result 4* -PR4-). After these homology pool enrichment steps, MSAs and HMMs were performed again for each, and HMMs were then compared against each other (pairwise) with HHalign [29]. Groups whose HMMs scored > 20 and did not contain two proteins from the same genome were combined. To detect remote orthologs from the generated sets, new HMMs were constructed, and a search in BPDB was conducted with the same considerations used to generate PR3 but with a more relaxed e-value (10) (*partial result 5* -PR5-). 

However, some genomic files may lack annotated putative genes, and so, the longest sequence was selected from each group of homologues and used as query in a tblastn on the excluded genomes. If a hit was detected in an annotated open reading frame (ORF), the corresponding protein was added to the orthologous group. If the hit was located in an unannotated region, the ORFFinder server was used to predict the longest ORF that included it, the novel protein was obtained by in silico translation and was then added to the corresponding homology groups, with the MSAs and HMMs repeated. In addition, a new expanded database was built on delta and gammabaculovirus genomes using ORFFinder, and a search was conducted on it with each of the HMMs. This generated a new batch of results (*partial result 6* -RP6-). Furthermore, HMMs were used as probes to search all potential ORFs present in gammabaculoviruses and deltabaculovirus with more relaxed criteria (e-value 10) to generate a new set of results (*partial result 7* -RP7-). Finally, those ortholog groups that were integrated by all alpha- and betabaculoviruses and missing all or some of the gammabaculoviruses or deltabaculovirus were analyzed by synteny or previously reported biological function, to generate the final result of the computational algorithm. For each of the partial results, InterProScan [30] was used to confirm correct grouping. Proteins not contained in any of the homologous groups (and not considered as paralogs after analyzing their similarity with established groups of orthologs) were considered derived from unique genes, and the total number of genes (shared and unique) present in *Baculoviridae* were counted. 

### 2.3. Orthologs Shared with Other Insect Viruses

Complete genomes of some viruses related to baculoviruses were retrieved from GenBank [9] (www.ncbi.nlm.nih.gov; last accesed on 1 June 2022) (Appendix A). A database containing all possible ORFs was generated using ORFFinder, and hypothetical proteins were generated from it. HMMs derived from orthologs found in baculoviruses by OBP application (only core genes) were used as probes to detect shared genes. Similarly, OBP was applied on the proteomes of these viruses from other viral families in a new database where the proteomes of each of the baculoviral prototypes were added (Autographa californica NPV -*Alphabaculovirus* Group I-, Lymantria dispar NPV -*Alphabaculovirus* Group II-, Cydia pomonella GV -*Betabaculovirus*-, Culex nigripalpus NPV -*Deltabaculovirus*-, Neodiprion lecontei NPV -*Gammabaculovirus*-) [10].

### 2.4. Baculoviral Phylogeny Inference

The set of 38 core proteins (plus additional ones detected as homologues in this work) from all considered baculoviruses (Appendix A) was obtained by applying OBP, aligning, concatenating and then using them to infer a phylogeny with the maximum likelihood method and the analysis “ModelFinder + tree reconstruction + ultrafast bootstrap (1000 replicates)” in IQ-TREE 2.1.2 [31]. The same phylogenetic analysis was carried out for the major occlusion body proteins of the prototype baculoviruses, including AcMNPV for *Alphabaculovirus* Group I, LdMNPV for *Alphabaculovirus* Group II, CpGV for *Betabaculovirus*, CuniNPV for *Deltabaculovirus* and NeleNPV for *Gammabaculovirus*, as well as other invertebrate viruses (Appendix A). Moreover, the nucleotide sequences corresponding to the ORFs of each core protein were retrieved from each genomic file using an ad hoc script. With this database, an evolutionary divergence analysis was performed in the MEGA11 [32] using the Kimura 2-parameters (K2P) model [33]. The criteria previously reported [16,34] were used as the only basis for grouping the baculoviral isolates considered in this study into different baculoviral species.

### 2.5. Protein Bioinformatic Studies

Secondary structure studies for polypeptide sequences were performed using JPred4 [35]. Hydrophobicity profiles were estimated using ProtScale [36], and protein disorder studied with PredictProtein (https://predictprotein.org/, last accesed on 30 November 2022) [37]. Sequence identity and similarity analyses were performed using an all-against-all strategy and pairwise alignments with Clustal Omega (default parameters). Identity was defined as the percentage of identical residues in the pairwise alignment, while similarity was calculated as the percentage of the sum of identities and similarities. For Z-score analyses [38,39], each of the baculoviral core protein sets [containing 297 natural protein sequences (natS)] were used to generate the corresponding sets of shuffled sequences (shfS) with the shuffle protein program of sequence manipulation suite [40] while preserving the individual length of each protein and its amino acid composition. Then, all natS were aligned among them and against the corresponding 297 shfS using the EMBOSS Needle pairwise sequence alignment program. Once the score average and the corresponding standard deviations from the alignments with shfS were obtained, the Z-scores were calculated using the following equation:Z-score = (SnatS − AshfS/natS)/s
where “SnatS” is the Score of each pairwise alignment of natS, “AshfS/natS” is the average scores obtained from pairwise alignments between each nats and each shfS, and “s” is the standard deviation of AshfS/natS scores.

## 3. Results

### 3.1. Baculovirus Genomes

Baculoviruses are a group of insect viruses with large cccdsDNA genomes of variable length (Figure 2A). This feature reveals that there were gains and losses of sequences throughout their evolution, including genome rearrangements of conserved sequences through inversion or translocation processes. These events resulted in the diversity of baculoviral species that are currently circulating on Earth.

Although most known genomes were between 125,000 and 135,000 bp, there was a wide range determined by a gammabaculovirus (81,755 bp; *Neodiprion lecontei NPV*) and a betabaculovirus (178,733 bp; *Xestia c-nigrum GV*). When genera were considered, the majority of alphabaculoviruses exhibited larger genomes than the others. For instance, isolates classified in group I of this genus had genomes that ranged from 111,953 bp (Maruca vitrata NPV) to 160,849 bp (*Orgyia pseudotsugata NPV*), while those of group II were between 105,555 bp (*Urbanus proteus NPV*) and 168,041 bp (*Leucania separata NPV*), betabaculoviruses were between 98,392 bp (*Diatraea saccharalis GV*) and 178,733 bp (*Xestia c-nigrum GV*) and gammabaculoviruses were between 81,755 bp (*Neodiprion lecontei NPV*) and 86,462 bp (*Neodiprion sertifer NPV*). The only known representative of deltabaculovirus, Culex nigripalpus NPV, had a genome of 108,252 bp, which is close to the average length for betabaculoviruses. The prototypes proposed for each of the genera (Autographa californica NPV -*Alphabaculovirus* Group I-, Lymantria dispar NPV -*Alphabaculovirus* Group II-, Cydia pomonella GV -*Betabaculovirus*-, Culex nigripalpus NPV -*Deltabaculovirus*-, Neodiprion lecontei NPV -*Gammabaculovirus*-) [10] had genomes with lengths that, when compared with each other, correlate with what was observed when considering all the genomes studied according to taxa.

When analyzing the genomic GC content (Figure 2B), alpha- and betabaculoviruses showed a similar distribution, while deltabaculovirus exhibited the highest value. However, some alphabaculoviruses from group I (e.g., *Orgyia pseudotsugata NPV* with 55%) and group II (e.g., *Lymantria dispar NPV* with 58%) showed the maximum GC contents. Although only three gammabaculovirus genomes were described, this genus contained the lowest percentage in GC since it ranged between 33% (*Neodiprion lecontei NPV*) and 34% (*Neodiprion sertifer NPV*). Gammabaculoviruses not only have the smallest genomes but are also the richest in AT, even though some alpha- and betabaculovirus genomes have similar GC percentages (e.g., *Oxyplax ochracea NPV* with 31% and *Cryptophlebia leucotreta GV* with 32%, respectively), making this not a unique property of that genus. Thus, the heterogeneity in baculovirus genomic lengths and richness in GC observed in this study and previously also reported makes it necessary to concentrate classificatory efforts on differential gene content.

### 3.2. Baculovirus Shared and Unique Protein Genes

Differences in the length of baculoviral genomes (as well as their biological and morphological properties) were correlated with the differential occurrence and number of protein-coding genes they possessed. This diversity in gene content was analyzed by an ad hoc automated computational algorithm composed of eight stages, named OBP (Figure 1), to detect groups of common ancestry that would allow the definition of the genetic signatures for the family and each of its genera. To achieve this, the accepted 38 core protein genes [17,18] were used as a calibration control set. This group of sequences, agreed upon by the scientific community as the shared core of baculoviral information [4,5], comprised genes that revealed both close and remote homology (including homologues for some in other invertebrate viral families, as will be corroborated later in this work). Therefore, it is an excellent learning set to characterize the entire *Baculoviridae* proteome based on their possible common ancestry. The OBP results showed the presence of 2412 different protein-coding sequences in *Baculoviridae*, confirmed the number of genes shared by all the genomes under study, and allowed for the discovery of those shared among some taxa as well as those potentially unique (Figure 3 and Appendix A). 

When not considering the core genes, the *Deltabaculovirus* genus (which, unfortunately for statistical purposes, was only represented by one member) shared the smallest amount of coding genome with the rest of the baculoviruses. In contrast, gammabaculoviruses, despite having smaller genomes and parasitizing hymenopterans, shared a significant number of sequences with alpha- and betabaculoviruses, both of which infect lepidopterans and are much more closely related.

The global analysis on the updated *Baculoviridae* genomic database did not reveal the loss of any accepted core gene, which demonstrates the success of the designed computational algorithm. However, it could suggest the incorporation of novel sequences not previously considered in this category due to their extremely remote homology (e.g., the gene that expresses the OB major protein, generally named polyhedrin/granulin) which require further inspection, as will be indicated later. In fact, most of the current 38 core genes could be detected in the early stages of OBP. Still, it is interesting to note that the latest sequences reported as ancestrally common (*ac53*, *ac78*, *ac101/p40*, *ac103/p48*, *ac110/pif7*) [17,18] could only be detected as orthologous for all the baculoviruses in stage 6 of OBP (Appendix A). Moreover, two other core genes (*desmoplakin* and *p6.9*) required the synteny analysis (stage 8 of OBP) to complete their presence in all baculoviral isolates considered, revealing that they are the sequences with shared ancestry that diverged the most through evolution. As in these two cases, another 16 orthologous clusters, previously not considered as baculovirus core genes, were detected, in which some or all the potential gammabaculovirus and deltabaculovirus homologues were missing [*polyhedrin*/*granulin* (*ac8*), *pk-1* (*ac10*), *f protein* (*ac23*), *dbp* or *DNA binding protein* (*ac25*), *v-ubiquitin* (*ac35*), *39k-pp31* (*ac36*), *lef11* (*ac37*), *nudix* (*ac38*), *fp25* (*ac61*), *ac106*/*107*, *ac75*, *ac76*, *p24* (*ac129*), *me53* (*ac139*), *chtb* (*ac145*), *ac146*]. Therefore, a thorough inspection for these sequences was carried out to either discard them as novel core genes or propose them in that category under the consideration of their remote orthologous condition. 

The case of *polyhedrin*/*granulin* gene is perhaps the most interesting, as orthologous sequences were detected in all members of alpha-, beta- and gammabaculoviruses, with only the deltabaculovirus homologue missing. However, there is a gene in CuniNPV that expresses the major OB protein, CUN085 [13,41]. Despite this, the synteny analysis, which considered *p6.9* and *desmoplakin* as shared genes in *Baculoviridae*, did not reveal a conserved genomic organization for CUN085 (Appendix A). Nevertheless, a sequence homology analysis did reveal protein regions with considerable sequential and structural conservation among the Polyhedrin/Granulin from alpha-, beta- and gammabaculoviruses and the CuniNPV major OB protein (Figure 4). Even the region containing the nuclear localization signal (NLS) and other residues considered important for the structuring of the protein were shown to be conserved [42,43].

The difference in CuniNPV polypeptide length, with respect to its potential orthologues in the other genera (882 amino acids vs. a median of 246 amino acids), was a characteristic observed in other core proteins of CuniNPV, such as Desmoplakin and Ac53. Moreover, this feature was also observed in other baculoviruses for Pif-7 and P6.9 (Appendix A). Additionally, similarity and identity analysis of CUN085 with its potential orthologous proteins showed values that were within the same range as other core proteins characterized by evidencing remote orthology, such as Desmoplakin (Figure 5). 

MOBP exhibited a high degree of conservation among baculoviruses infecting Lepidoptera and Hymenoptera, with a greater divergence observed in deltabaculovirus. However, a robust phylogeny could even be inferred among the homologues of each baculoviral prototype and the proteins of other invertebrate viruses that also embed their virions in crystalline matrices (Figure 6).

Likewise, a Z-score analysis revealed values comparable to those obtained when studying other accepted core genes in *Baculoviridae* (Appendix A). Although MOBP presented some Z-score values below zero, Desmoplakin and P6.9 (accepted orthologous proteins) showed similar behavior, especially in the case of CuniNPV Z-scores. In the case of Desmoplakin, 49 alphabaculovirus, 74 betabaculovirus and three gammabaculovirus Z-score values were below zero. For P6.9, 180 alphabaculovirus, 70 betabaculovirus and two gammabaculovirus Z-score values below zero were obtained. However, for MOBP, only two alphabaculovirus, 14 betabaculovirus and one gammabaculovirus Z-score values were below zero. Therefore, by functional homology (i.e., protein that constitutes the paracrystalline matrix of OB) and partial sequential homology, it is appropriate to consider that the “major occlusion body protein” (MOBP) in baculoviruses should be postulated as a product of the 39th core gene. In fact, this would be consistent with a morphological characteristic of the virions of this viral family (occlusion bodies) that is common to all the known members.

The remaining sequences with homologues not detected in gammabaculoviruses and/or deltabaculovirus did not meet the criteria for synteny, and no proteins with similar biological functions were reported, which would allow for a similarity study like the one conducted with MOBP. Specifically, no functional or significant sequential homologues of *pk-1* (not detected in delta- and gammabaculoviruses), *f protein* (not detected in gammabaculoviruses), *dbp* (not detected in deltabaculovirus), *v-ubiquitin* (not detected in delta- and gammabaculoviruses), *39k-pp31* (not detected in delta- and gammabaculoviruses), *lef11* (not detected in deltabaculovirus), *nudix* (not detected in delta- and gammabaculoviruses), *fp25* (not detected in delta- and gammabaculoviruses), *ac106*/*107* (not detected in deltabaculovirus), *ac75* (not detected in deltabaculovirus), *ac76* (not detected in deltabaculovirus), *p24* (not detected in delta- and gammabaculoviruses), *me53* (not detected in delta- and gammabaculoviruses), *chtb* (not detected in deltabaculovirus) and *ac146* (not detected in deltabaculovirus) were found in the baculoviruses that were not part of the orthologous groups. It should be noted that a partial similarity with CUN076 was found for Lef11 protein, and a similar situation occurred with Ac106/107 and CUN053 and between Ac131 and CUN103 (Appendix A). However, these homology findings must be refined and functional assays should be added for their definitive incorporation as the 40th, 41st and 42nd core genes, respectively. Regardless, this set of protein-coding genes should be considered as the next group of candidate sequences where novel core genes could be postulated. 

In addition, the computational algorithm was applied to assess the homology of the 38 previously accepted baculovirus core genes in other invertebrate virus genomes (Appendix A). This analysis allowed for the detection of sequences with potential common ancestry for 15 of the 38 core genes, as detailed in Appendix A. This does not imply that other proteins from these viruses cannot be linked to baculovirus core genes in other studies by other homology search strategies. Undoubtedly, the lack of similarity and syntenic conservation makes it difficult to identify them. Interestingly, among the 15 shared genes, there were seven *pif* (*pif0*, *pif1*, *pif2*, *pif3*, *pif4*, *pif5*, *pif7*) in some of the non-baculoviral viral genomes. Additionally, homologues to four core baculoviral genes (*dna helicase*, *pif0*, *pif2* and *pif3*) were found in all the genomes studied.

On the other hand, the analysis of common ancestry in baculovirus coding sequences revealed the existence of numerous potentially unique genes (ORFs with no apparent similarity to other sequences in the same dataset), as well as others that multiplied through copies within some genomes (paralog genes). Regarding this, protein-coding genes that are represented only in single baculoviral isolates comprise a set of 1740 sequences (Appendix A). It is possible that OBP did not detect remote homologies in other members, or that some of them do not express functional proteins despite being detected as ORFs. This group of putative protein genes should be explored in greater depth to determine their ancestral roots or their origins de novo, and to assign biological roles in the viral cycle. Meanwhile, the situation with paralogous genes is clearer since many of them were detected as expressed proteins or received a function. For example, the genes *bro* (*baculovirus repeated orfs*), *dbp* (*dna binding protein*), *helicase*, *iap* (*inhibitor of apoptosis*), *odv-e66* and *p26*, among others (Appendix A), fall under this group.

The results obtained from the application of OBP on 297 baculoviral genomes also allowed the verification of genes that are ancestrally shared between genera (Appendix A) and to obtain the genetic content of each virus (Appendix A). Thus, the number of protein genes in different genera showed that, on average, alphabaculoviruses expressed a greater number of proteins than other taxa. In contrast, gammabaculoviruses (according to their genomic dimensions) had the lowest gene content.

In accordance with the number of annotated protein genes in the studied genomes (including paralogs), the gene content ranged from 124 to 161 in alphabaculoviruses group I (Oxyplax ochracea NPV and Dasychira pudibunda NPV, respectively), 102 to 178 in alphabaculoviruses group II (Ectropis obliqua NPV and Lymantria dispar NPV, respectively), 116 to 183 in betabaculoviruses (Choristoneura fumiferana GV and Pseudaletia unipuncta GV, respectively), 109 in deltabaculovirus (Culex nigripalpus NPV) and 89 to 93 in gammabaculovirus (Neodiprion lecontei NPV and Neodiprion abietis NPV, respectively). Therefore, the limits in protein gene content for *Baculoviridae* were 89 and 183 (Neodiprion lecontei NPV and Pseudaletia unipuncta GV, respectively), with the consideration that these values may include hypothetical genes that may not be transcriptional and translational units and their biological functions occurs as non-coding regions of DNA. 

### 3.3. Baculovirus Phylogeny

The phylogenetic inference carried out on the set of genomes selected in this study clearly revealed the baculovirus distribution in the four main taxa (*Alpha*-, *Beta*-, *Gamma*- and *Deltabaculovirus* genera) including the proper separation of alphabaculoviruses into groups I and 2. This result can be achieved by using the 38 accepted core proteins, plus MOBP as the 39th core protein (Figure 7).

As established in previous reports, the current main criterion for classifying baculoviral isolates into species relies on the genetic distances derived from comparing the ancestrally common genomic region (38 core protein genes) [16]. In this sense, the metagenome of baculoviruses known on Earth was studied in terms of the genetic distances based on K2P (Figure 8).

The results showed that the median of the genetic distances within each genus (including comparison between group I and group II alphabaculoviruses) was less than 1, while they were significantly higher than this value when the comparison between genera was made. This reinforces what was observed in the phylogenetic inference (Figure 7) and when the protein-coding genome with common ancestry was studied (Figure 3), showing that the taxa used to classify baculoviruses were adequate and responded to their evolutionary history, with the unique deltabaculovirus being the most distant representative of *Baculoviridae*.

Regarding all K2P distances between pairs of genomes as the only classification criteria (Appendix A), the 74 isolates belonging to group I of alphabaculoviruses were distributed in 24 different species, while the 144 isolates classified within group II of alphabaculoviruses were divided into 52 species. Meanwhile, the 75 considered betabaculoviruses were classified into 26 species and the three gammabaculoviruses corresponded to three different species. Considering the only species belonging to deltabaculovirus (CuniNPV), until June 2022, there would be 107 baculoviral species from the analysis of 297 isolates with completely sequenced genomes. It is worth mentioning that isolates with low K2P distances were, therefore, considered to belong to the same species and may be accepted as different because they had differential gene content, virion morphology or significantly different biological features (e.g., pathogenicity and virulence in different hosts). Additionally, the organization of baculoviral genomes based on K2P values reproduced the same clustering as that observed by phylogenetic inference (Appendix A). When the analyses of length, GC content and gene numbers were repeated for the total of 107 genomes from different species (without considering isolates from the same species) (Appendix A), the trends observed were equivalent to those shown above (Figure 2 and Appendix A).

## 4. Discussion

Biological entities are constituted as carbon-based supra-macromolecular organizations that use the central dogma of information (from nucleic acids to proteins), such as prokaryotes, eukaryotes, their organelles and viruses. They present ancestral relationships whose divergence can be caused by time (reproduction events) and by different environmental pressures (biotic and abiotic) that they went through without becoming extinct. Within the life sciences, ancestors such as LUCA (last universal common ancestor) or LECA (last eukaryotic common ancestors) [44] appear as possible reconstructions and inferences based on current and past biological diversity, where the fossil record (when it exists) plays a prominent role. In viruses, such interpretations are much more complex. However, the analysis of the type of genomic nucleic acid and the mechanisms by which it is expressed (Baltimore classification), its sequence characteristics (e.g., length, GC content, presence of nucleic acid structures), its gene content with the sequential and/or functional similarity relationships (concepts such as orthology/homology and paralogy) and its organization (synteny) result in excellent objectifiable characteristics to group them taxonomically and speculate about their common evolutionary histories [45,46].

From the first descriptions of baculoviruses to the current diversity of members reported with completely sequenced genomes, there were changes in their considerations and classification [4,5]. It is now possible to establish that these arthropod-viruses are related to other representatives that also infect those hosts, and more distantly to other viruses that also carry large dsDNA genomes [46,47] as recognized in the new taxonomic groupings formed by the order *Lefavirales* within the class *Naldaviricetes* because they are ancestrally related nuclear arthropod large DNA viruses (NALDVs) [48,49]. The exponential increase in the availability of genomic sequences and the improvement in the syntactic knowledge of the functional elements of nucleic acids showed a major challenge for bioinformatics and the search for homology and ancestry among sequences from different entities [50]. Under these considerations, in this work, the relationships among protein-coding genes (annotated and not annotated by the authors who described them) were reviewed using powerful bioinformatics tools. This allowed us to reconsider which are the protein coding sequences shared by all baculoviruses, being able to recognize that the 38 previously reported were properly included in that category [17,18]. In addition, it was possible to detect another subset of sequences close to entering in this common ancestral group, such as the gene that expresses the major protein of OBs, a distinctive characteristic of *Baculoviridae*, named in this report as *mobp* (*major occlusion body protein* gene) considering its variants *polyhedrin* (in alpha- and gammabaculoviruses), *granulin* (in betabaculoviruses) and *cun085* (in deltabaculovirus). Although *cun085* does not have a syntenic organization similar to that occurring in other baculoviruses and encode a longer polypeptide than its homologs (characteristics that other core proteins share), it does have numerous amino acids-conserved regions that can integrate it at the same ortholog group that contain polyhedrins and granulins. In this sense, the gene that expresses the major occlusion body protein could be considered as the 39th core gene of *Baculoviridae*. Its incorporation does not compromise phylogenetic inference and allows linking baculoviruses with other invertebrate viruses, showing that both the PIFs [7] and the major structural virion protein appeared in nature millions of years ago, later dividing into numerous successful branches of invertebrate viruses carrying large dsDNA genomes. MOBP comprises a set of ancestrally related polypeptides according to our results, whose constitutive genetic elements are extremely interesting, given the amount of protein mass they generate [51]. From the beginnings of the description of polyhedrin proteins expressed by NPVs that infect Lepidoptera (now taxonomically classified as alphabaculoviruses) and granulin proteins of the GVs (now taxonomically classified as betabaculoviruses), biochemical, structural and functional relationships between them were established [34,52,53]. Subsequently, the molecular description of the baculoviruses that infect Hymenoptera (now taxonomically classified as gammabaculoviruses) showed that the gene expressing the major protein of the OBs was also related to the previous ones [11,54,55]. Even studies carried out on other invertebrate dsDNA viruses that generate an occluded morphotype of their virions revealed a common ancestry for the structural proteins responsible for that biological structure [56,57,58]. Although virion occlusion is a strategy also used by RNA viruses that infect invertebrates, and presenting some shared biochemical characteristics, the ancestry with DNA viruses is not yet supported [42,43]. For all these reasons, it was striking that the only known representative of dipteran-infecting NPV, a baculovirus that also shows an occluded phenotype, expressed a protein ancestrally unrelated to invertebrate DNA viruses to generate OBs [13,41]. The exhaustive analysis carried out in this work was able to find the regions of homology for the protein that occludes virions in all the representatives with sequenced genomes of the four genera of *Baculoviridae*. This common ancestry among Cuni085, polyhedrins and granulins seems remote because the deltabaculovirus ortholog probably underwent an evolutionary process that would allow the OBs to sustain themselves in aquatic environments, different pressure than occurred with the baculoviruses that infect Lepidoptera and Hymenoptera.

Since there are other ortholog groups where only the CuniNPV homologous protein is missing, it is possible that other core genes may be added in the future as search engines for similar protein patterns and the functional study of each baculoviral polypeptide improve. Additionally, it may happen that the description of more deltabaculoviruses affects the consensus of core genes accepted up to now. These will be the conditions to change the ancestrally common coding core of *Baculoviridae*. Particularly, it will be necessary to inspect more thoroughly if CUN053, CUN076 and CUN103 proteins have a common ancestry with Ac106 (a nuclear protein that modulate transcriptional processes) [59], Lef11 (a nuclear protein involved in baculoviral DNA replication) [60,61] and Ac131 (a polyhedron envelope/calyx protein) [4], respectively. The activities of these polypeptides seem to be important in the infection cycle, and so, it is possible that the sequences detected in the CuniNPV genome are also part of the preserved genomic fraction of *Baculoviridae*.

It is also interesting to note the high diversity of protein-coding genes carried by baculoviruses (2412 putative different ORFs). Undoubtedly, dsDNA viruses evolve primarily through the gain and loss of new gene functions; and viruses, transposons and endosymbiont bacteria appear to be central tools in invertebrate gene flow [62]. The viral species concept is probably a human construct rather than a natural property that only needs to be observed and described, as is the case with eukaryotic species. Nonetheless, the genetic content and genetic distances of the shared sequences provide a suitable basis for understanding the Earth’s virome. 

## 5. Conclusions

The identification of common characteristics that respond to measurable features in biological entities aids in the determination of criteria for their classification. This is true not only for organisms but also for viruses, where classical taxonomic definitions are in tension and where the fossil record is almost non-existent. This study focused on an exhaustive analysis of the protein-coding genomic fraction of isolates with fully sequenced genomes reported as members of *Baculoviridae*, with the aim of identifying common genetic characteristics by taxa. Baculovirus genomes are single cccdsDNA of varying length (81,755 bp–178,733 bp), GC proportion (31–58%) and protein-gene content (89–183 ORFs), The shared gene fraction (at least 39 ORFs) and evolutionary distances from the common region allow them to be grouped into five well-differentiated taxa [recognized as genera *Alphabaculovirus* (group I and group II), *Betabaculovirus*, *Deltabaculovirus* and *Gammabaculovirus*] and into 107 species (until June 2022).

These findings will help to better understand the evolutionary history of this viral family, given the current diversity of described representatives throughout the entire planet. Moreover, these findings will contribute to the genomic annotation of new baculoviral isolates, and the design of functional genomics experiments to move towards gene ontology studies.

## Figures and Tables

**Figure 1 viruses-15-01091-f001:**
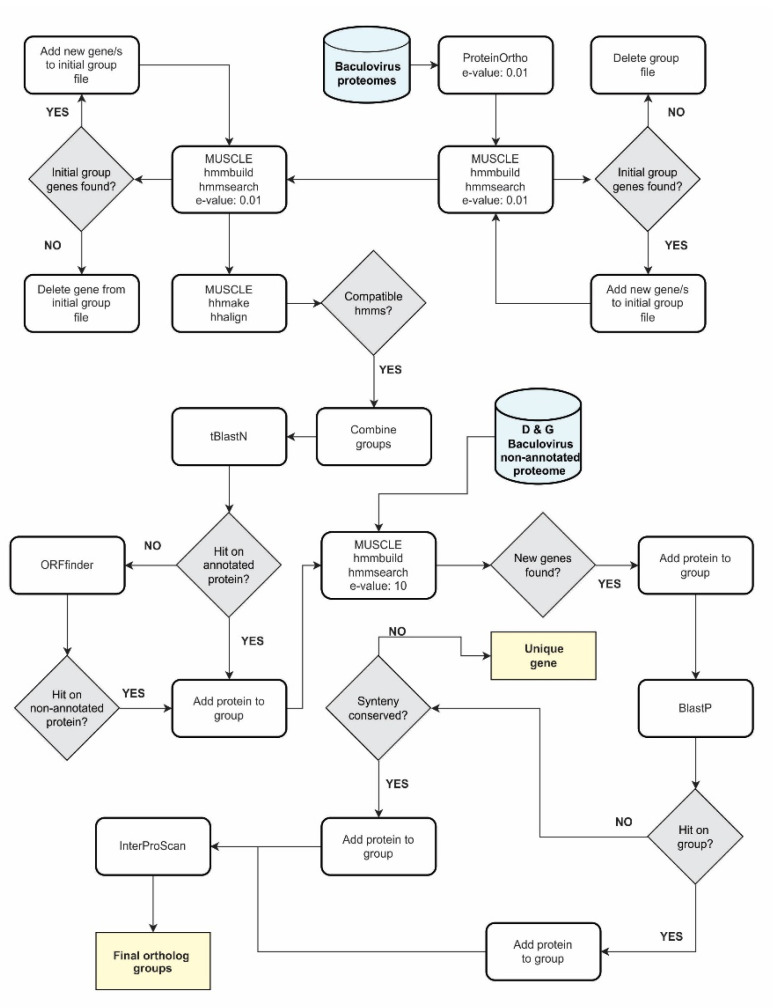
Ortholog Bioinformatic Pipeline. The diagram illustrates the automated computational algorithm for identifying orthologous gene clusters in baculoviruses with sequenced genomes. The expected final results are shown in yellow boxes, which include homologous genes present in more than 1 genome (ortholog groups) or unique genes (with no detectable homology in another baculoviral genome). From the pool of potentially unique genes, paralogous genes were identified by comparing them against the already detected ortholog clusters. The databases used are indicated in light blue (“D & G Baculovirus non-annotated proteome” refers to the possible proteome constructed by in silico translation from deltabaculovirus and gammabaculovirus genomes).

**Figure 2 viruses-15-01091-f002:**
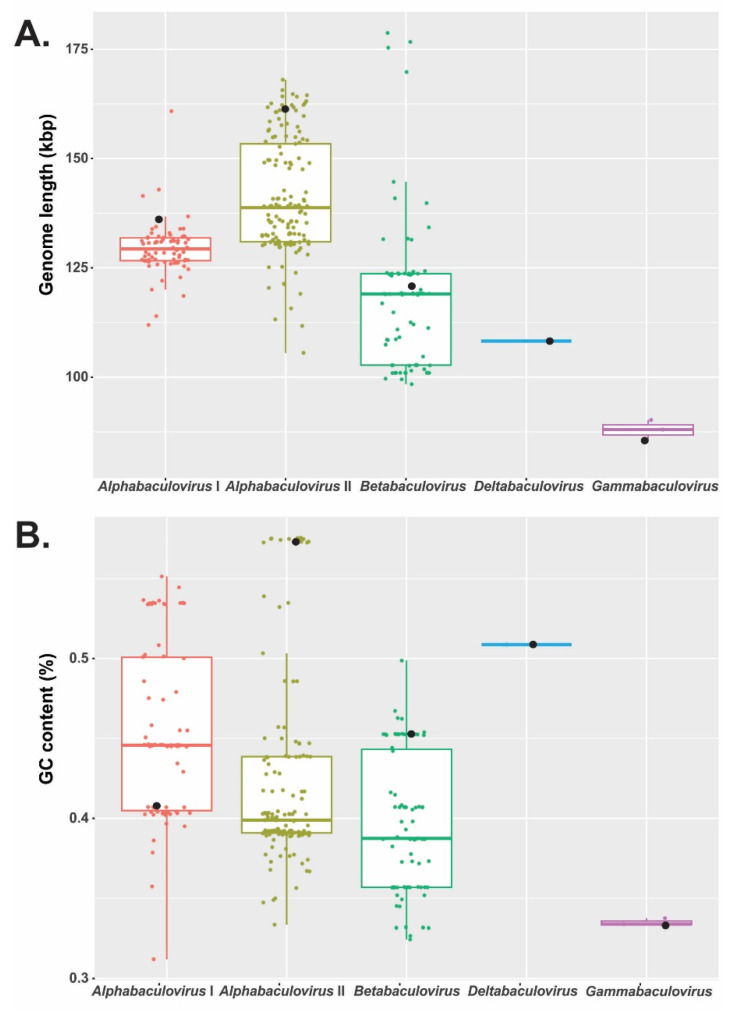
Length and GC content of baculoviral genomes. (**A**) Box plot of the 297 baculoviral genomes according to their length (ordinate axis) and considering taxa (abscissa axis). (**B**) Box plot of the 297 baculoviral genomes according to their GC content (ordinate axis) and considering taxa (abscissa axis). Each dot represents the datum for each genome. The value linked to each of the baculovirus prototypes (AcMNPV for *Alphabaculovirus* Group I; LdMNPV for *Alphabaculovirus* Group II; CpGV for *Betabaculovirus*; CuniNPV for *Deltabaculovirus*; NeleNPV for *Gammabaculovirus*) is indicated with black circles.

**Figure 3 viruses-15-01091-f003:**
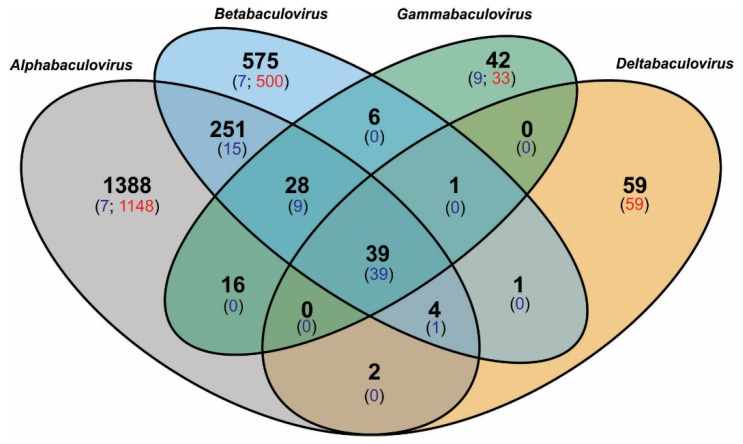
Protein-genes in *Baculoviridae*. The Venn diagram shows the numbers of all putative baculoviral protein-coding genes distributed by genera and indicating which ones are shared among taxa or are unique. The detail of the identity of each gene is shown in Appendix A. These results derive from the application of the automated algorithm generated in this work (OBP) to determine ortholog groups and putative unique genes. The numbers in black bold letters represent amounts of shared protein genes. In regions without overlapping ellipses the numbers in black bold letters also include unique genes. It is indicated with numbers in blue letters between parentheses how many of the shared protein genes form groups of orthologs in all viruses considered, and in red letters the amount of putative unique genes per genus. To facilitate the understanding of this figure, examples of interpretation are provided: (a) the genomes of alpha and betabaculoviruses have 251 protein-coding genes (bold black number), each of which is present in at least one alphabaculovirus and one betabaculovirus but not in any members of other genera; 15 (blue number) of these 251 genes have homologs in all the alpha and betabaculovirus genomes analyzed. (b) Alphabaculoviruses possess 1388 (bold black number) protein-coding genes that lack homologs in any other baculoviral genus; of these, 7 (blue number) have homologs in all the alphabaculovirus genomes analyzed and 1148 (red number) are unique to a single genome. This graph does not include paralog protein genes.

**Figure 4 viruses-15-01091-f004:**
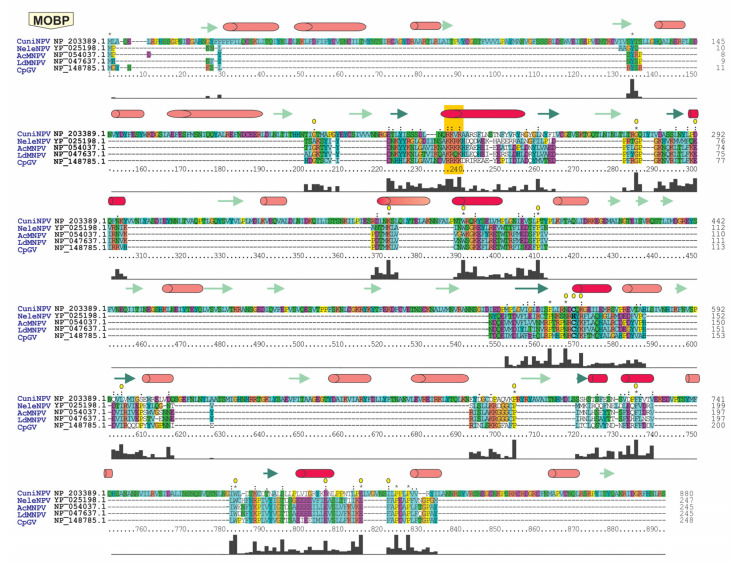
Baculovirus major occlusion body protein. The multiple sequence alignment (MSA) of the major occlusion body protein (MOBP) for baculoviral prototypes (AcMNPV for *Alphabaculovirus* Group I, LdMNPV for *Alphabaculovirus* Group II, CpGV for *Betabaculovirus*, CuniNPV for *Deltabaculovirus* and NeleNPV for *Gammabaculovirus*) is shown. The MSA was performed with all MOBPs from the baculoviral 297 genomes studied, but only the prototypes are shown. The MSA was created with Clustal Omega, and then a manual revision was carried out considering the structural features predicted with Jpred. The alpha helices are shown in red, the beta sheets in green (with those conserved in all 5 proteins in dark color and those present only in CuniNPV in light color) and the conserved nuclear localization signal (NLS) in orange (MSA position 240). Position 570 of the MSA, where cysteines (and histidines in the case of gammabaculoviruses) are conserved, is indicated in bold. These residues are involved in the formation of the disulfide bridge that holds together two MOBP trimers. In addition, the residues that are conserved in the MSA and were determined to be important in previous structural analyses of alpha- and betabaculovirus MOBPs [42,43] are highlighted with yellow circles above the corresponding amino acid position. Asterisks (*) indicate positions with identity, while points (.) and colons (:) indicate positions with conservative changes. Below the multiple alignment, the MSA quality curve is displayed.

**Figure 5 viruses-15-01091-f005:**
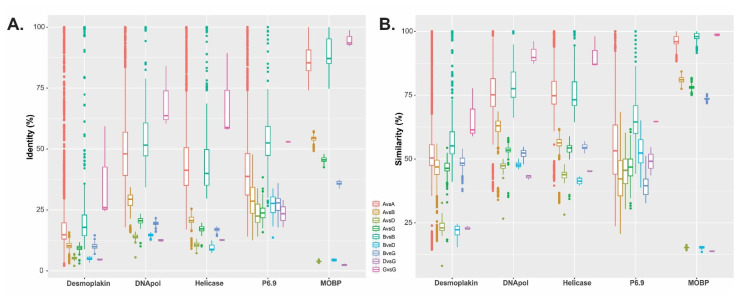
Similarity and identity study for MOBP. Box plots showing Identity (**A**) and Similarity (**B**) values for some baculovirus core proteins (Desmoplakin, DNA polymerase, DNA Helicase, P6.9 and the major occlusion body protein -MOBP-) among isolates from different genera. The box plots include outliers and the abbreviations A, B, D and G stand for *Alphabaculovirus*, *Betabaculovirus*, *Deltabaculovirus* and *Gammabaculovirus*, respectively.

**Figure 6 viruses-15-01091-f006:**
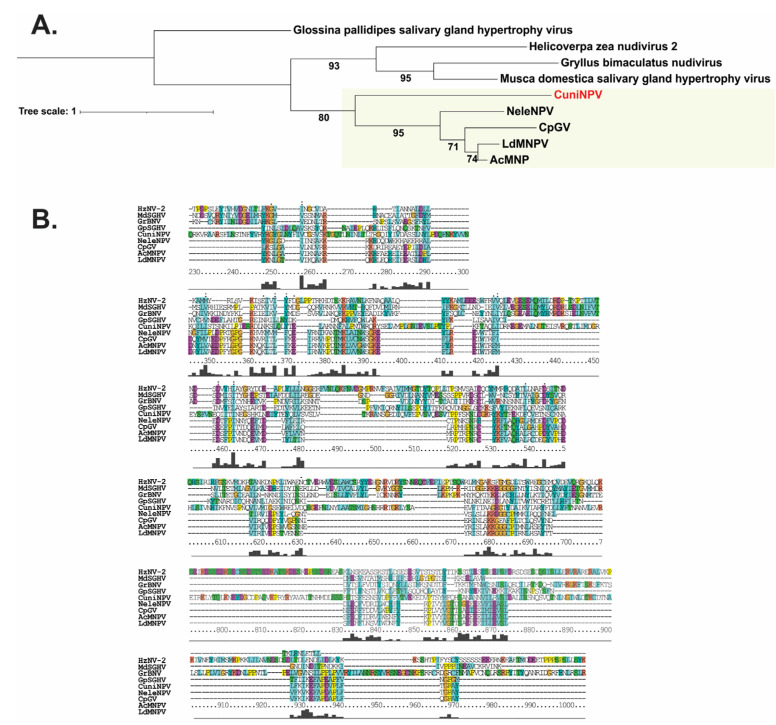
Phylogeny inference based on major occlusion body protein. (**A**) Phylogram based on the major occlusion body protein (MOBP) of baculovirus prototypes (AcMNPV for *Alphabaculovirus* Group I; LdMNPV for *Alphabaculovirus* Group II; CpGV for *Betabaculovirus*; CuniNPV for *Deltabaculovirus*; NeleNPV for *Gammabaculovirus*) and other invertebrate viruses [YP_001111332.1 (Gryllus bimaculatus nudivirus); YP_004956818.1 (Helicoverpa zea nudivirus 2); YP_001687041.1 (Glossina pallidipes salivary gland hypertrophy virus); YP_001883404.1 (Musca domestica salivary gland hypertrophy virus)]. The colored region contains the main taxa of *Baculoviridae*. CuniNPV (CUN085) is indicated in red letters. Bootstrap values are indicated. (**B**) Some relevant parts of the multiple sequence alignment used for the MOBP phylogenetic analysis.

**Figure 7 viruses-15-01091-f007:**
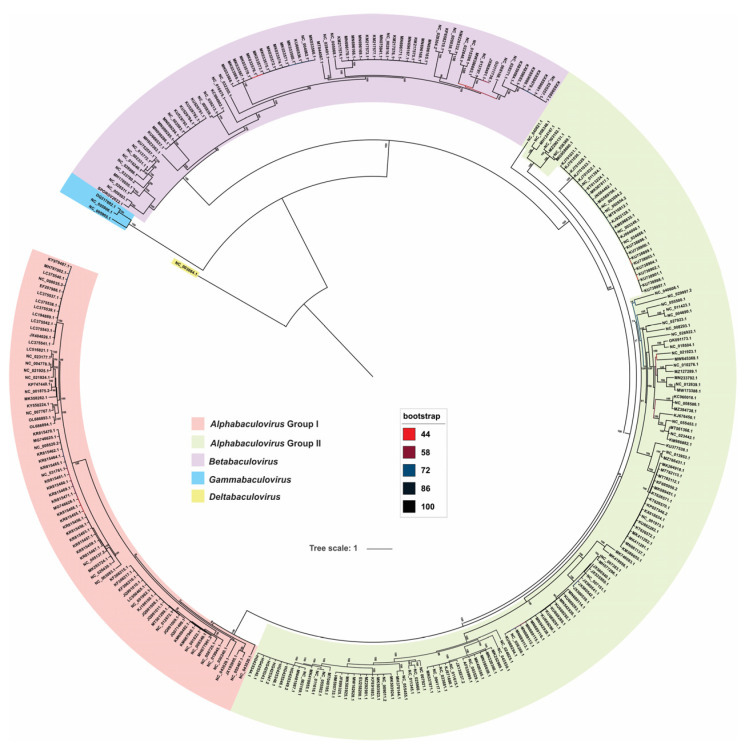
Baculovirus phylogeny. Phylogram based on a concatemer built with the 38 baculovirus core proteins plus MOBP (major occlusion body protein, also named polyhedrin or granulin). Baculoviral isolates are indicated by their GenBank accession number. The colored regions represent the main taxa of *Baculoviridae*. Bootstrap values are indicated and represented with a color scale which references are included.

**Figure 8 viruses-15-01091-f008:**
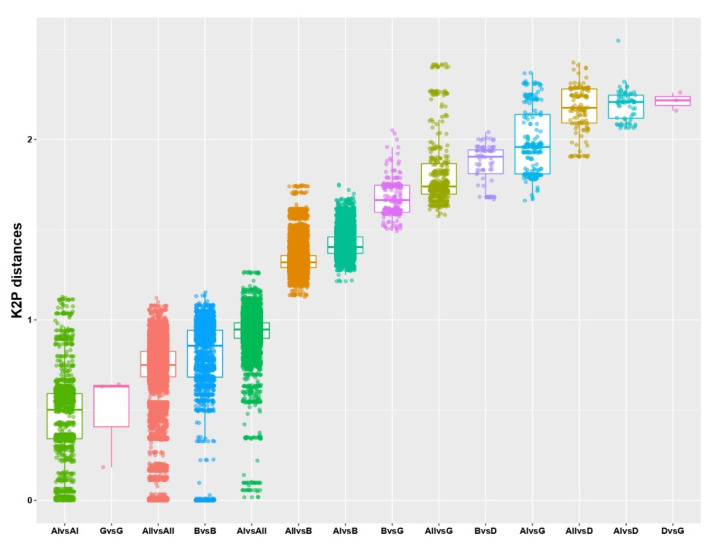
Baculovirus core gene distances. Box plot showing the K2P (Kimura 2-parameters) distances (ordinate axis) between taxa (abscissa axis). K2P distances were estimated from the concatenation of the ORFs corresponding to the 38 core proteins shared by all baculoviruses. Comparisons between taxa are shown in order from closest to largest based on medians. A1: alphabaculoviruses of Group I; A2: alphabaculoviruses of Group I; B: betabaculoviruses; D: deltabaculovirus; G: gammabaculoviruses.

## Data Availability

Not applicable.

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
