# Peer review of "Protein-Gene Orthology in Baculoviridae: An Exhaustive Analysis to Redefine the Ancestrally Common Coding Sequences"

_viruses, 2023, doi:10.3390/v15051091_

Round 1

Reviewer 1 Report

This is a well written manuscript on the gene content, orthology and phylogeny of baculoviruses based on 250 complete genomes availabe at Genebank. Though the analyses are well presented and interesting, the novelty of conclusions is limited and does not go well beyond previous baculovirus genome comparisons with a smaller data set.

Genome length, GC content, shared protein genes, gene contents, core gene based phylogeny, etc. have been reported manifoldly in other studies. Not surprisingly, the analyses confirmed the existance of 38 baculovirus core genes which was previously established. I would assume that if one of these core genes was missing from a baculovirus genome, it would have been reported previously.

The only novel hypothesis of the manuscriupt, i.e. the proposed homology of CUN085 with Polyhedrin/Granulin, is not well substantiated and mainly moved to Appendix A. However, presentation of this part is poor and I did not understand on which basis this homology was detected. The presented sequence logos do not contribute to understand the proposed homology. An alignment would have been more suitable than the logos that are biased to the overwhelming majority of alpha-, beta-, and gammabaculoviruses in the comparison.

The authors should consider to rewrite the manuscript as a review rather than a research paper and provide a more elaborated analysis to support their hypothesis of a common ancestry of baculovirus MOBPs.  Providing clear evidence on this issue would justify a paper on its own. 

Author Response

Reviewer #1

Comments and Suggestions for Authors

This is a well written manuscript on the gene content, orthology and phylogeny of baculoviruses based on 250 complete genomes availabe at Genebank. Though the analyses are well presented and interesting, the novelty of conclusions is limited and does not go well beyond previous baculovirus genome comparisons with a smaller data set.

Genome length, GC content, shared protein genes, gene contents, core gene based phylogeny, etc. have been reported manifoldly in other studies. Not surprisingly, the analyses confirmed the existance of 38 baculovirus core genes which was previously established. I would assume that if one of these core genes was missing from a baculovirus genome, it would have been reported previously.

We appreciate the reviewer's comments on our manuscript. It is true that some of the analyses presented have been reported before, but the growing number of available baculoviral genomes warranted further investigation. Therefore, we not only examined gene orthology, but also reanalyzed typical genomic features. Our study proposes a new and comprehensive bioinformatics pipeline, which could be useful for other viral families. Furthermore, the execution of this computational procedure allowed us to demonstrate that the 38 core genes of baculoviruses had been properly selected, as no similar procedure existed in the literature that would have enabled the identification of this complete set. In addition to this novelty, we have identified other potential CuniNPV genes that could be part of new core genes. Among them, we provide evidence that Cun085 (sequence encoding the major protein of the deltabaculovirus occlusion body) has homology with the other baculoviral genes that express polyhedrin and granulin. The number of core genes in Baculoviridae is not a settled question, and we believe that it is still possible to increase this number. In this regard, our study proposes other candidates (Ac106, Lef11 and Ac131), with a focus on the major protein of the occlusion body, given its previous experimental evidence.

The only novel hypothesis of the manuscriupt, i.e. the proposed homology of CUN085 with Polyhedrin/Granulin, is not well substantiated and mainly moved to Appendix A. However, presentation of this part is poor and I did not understand on which basis this homology was detected. The presented sequence logos do not contribute to understand the proposed homology. An alignment would have been more suitable than the logos that are biased to the overwhelming majority of alpha-, beta-, and gammabaculoviruses in the comparison.

We appreciate the reviewer's feedback and agree that the main focus of our work is to propose a new core gene, and as such, the main evidence should be presented in the main body of the manuscript. In response to this comment, we have reorganized the manuscript by removing Appendix A and incorporating the main results into the main text. Additionally, to further support our findings regarding the homology between polyhedrins, granulins, and Cun085, we have included new results that we hope will address any remaining concerns raised by the reviewer.

The authors should consider to rewrite the manuscript as a review rather than a research paper and provide a more elaborated analysis to support their hypothesis of a common ancestry of baculovirus MOBPs.  Providing clear evidence on this issue would justify a paper on its own. 

 We appreciate the reviewer's concern and have taken steps to provide additional evidence to more strongly support our hypothesis that MOBP is a new core gene. Given the extensive research we have conducted and the novelty of some of our findings, we believe that our manuscript is suitable as an original research paper. The notion that MOBP is an ancestrally conserved protein in Baculoviridae is believed to be well-supported by the new analyses that we have introduced in the text. We also emphasize the significance of the computational algorithm that we have developed, as it is the first of its kind that is capable of detecting all the core proteins of this viral family. Furthermore, we have identified other potential proteins in CuniNPV that could be part of new core gene groups. We believe that these findings are also original and offer new insights into baculovirus genetics.

Reviewer 2 Report

1. The data presentation needs to be improved. Figure 1 and Figure 2 both need re-draw and increase the resolution so that they are readable.

2. Minor improvements are needed for  English grammar or typos.

Author Response

Reviewer #2

Comments and Suggestions for Authors

  1. The data presentation needs to be improved. Figure 1 and Figure 2 both need re-draw and increase the resolution so that they are readable.
  2. Minor improvements are needed for English grammar or typos.

We appreciate the comments made by the reviewer. Consequently, we have improved the resolution of the figures, improved the presentation of our results, and have performed an extensive revision of the English language. We trust that this revised version of the manuscript provides improved quality and clarity in our exposition and justification of our findings.

Reviewer 3 Report

In this manuscript, the authors conducted extensive bioinformical analyses on all baculoviral protein-coding sequences from 250 genomes, and identified polyhedron/granulin/cuni085 as the 39th core gene among baculoviruses. Although the novelty of the report is limited, it does provide useful information about the genome organization and evolution of baculoviruses. Certain issues need to be addressed before the manuscript is suitable for publication: 1) The manuscript is not well organized. There is lots of not so useful information in the manuscript. Carrying too much such information will distract attention of the readers. 2) Some of the basic introductions about baculovirus are not accurate, need to be modified. 3) Some of the detail data need to be carefully checked. Please see the following comments.

1)      Appendix A, most of the information in Appendix A is not necessary and the brief description in main text is fine. I suggest removing the Appendix A from the paper, but 1) retaining Appendix-Figure 5 as a new figure, and provide the alignment used for constructing the phylogeny tree as a supplementary data; 2) retaining Table Appendix A and making it a new Supplementary table.

2)      Fig. 2A is not necessary. Fig. 2B, it is better to use kb, and please enlarge the font so it is readable.

3)      Figure S2 is not necessary.

4)      Fig. 4 legend: it is difficult to understand the following sentence “Between parentheses (in black letters) it is indicated how many of the shared protein genes form groups of orthologs with sequences present in all viruses considered”, and I do not understand the related numbers in Figure 4.

5)      Figure 5 and related description are not necessary.

6)      Table S2 is not necessary; it can be easily included in Table S4.

7)      Table S4. It is better to put SFHV together instead of separate them in the table. I did not check the data in detail, but find that P33 homologues should be present in most of the invertebrate viruses, but it was N in the table. Please check the data carefully.

8)      Result 3.3, I found the entire section unnecessary, as baculovirus phylogeny have been extensively studied, and this study did not add any new information.

9)      Abstract, “more than 250 genome”, change to “nearly 300 genomes”

10)   Introduction: Our knowledge of the two phenotypes of baculovirus, ie. BV and ODV, are largely generated from lepidopteran baculoviruses (alphabaculovirus and betabaculovirus). Currently it is generally believed that gammabaculoviruses do not contain a BV life cycle, and it remains unknown if deltabaculovirus contains BV. Please modify the related description in the manuscript, such as line 33-36.

11)   Introduction: Line 39-40. “OBs may contain 1 or more ODVs, and these in turn may contain 1 or more nucleocapsids depending on the baculoviral species”. This is not a correct description, as GV normally only contain 1 ODV.

12)   Introduction: line 74-75:” the number of homologous genes (known as core genes)”, this definition is incorrect. Core genes are the genes that are conserved in all the baculovirus genomes. While homologous genes are the ones that can find homologues in other baculovirus genomes. Please correct.

13)   Line 238, change “wide limit” to “wide range”

14)   Line 241, delete “thus”

15)   Line 255, “the other ones have greater in AT in similar values”, please revise the sentence to make it easier to understand

16)   Line 276-277 please remove “probably derived from the coevolution of these viruses with arthropods and their expansion throughout the entire planet”, as it is more suitable in the discussion section than in the result section.

17)   Line 356, what is “delta- y gammabaculoviruses”?

18)   Discussion, the current discussion is quite long and by removing the phylogeny results, the discussion needs to be revised.

19)   The English writing needs to be improved.

Author Response

Reviewer #3

Comments and Suggestions for Authors

In this manuscript, the authors conducted extensive bioinformical analyses on all baculoviral protein-coding sequences from 250 genomes, and identified polyhedron/granulin/cuni085 as the 39th core gene among baculoviruses. Although the novelty of the report is limited, it does provide useful information about the genome organization and evolution of baculoviruses. Certain issues need to be addressed before the manuscript is suitable for publication: 1) The manuscript is not well organized. There is lots of not so useful information in the manuscript. Carrying too much such information will distract attention of the readers. 2) Some of the basic introductions about baculovirus are not accurate, need to be modified. 3) Some of the detail data need to be carefully checked. Please see the following comments.

 We appreciate the comments and suggestions made by the reviewer. Below we respond to each of the detailed points. As a general matter, based on such observations, we have carried out a restructuring of the manuscript.

1) Appendix A, most of the information in Appendix A is not necessary and the brief description in main text is fine. I suggest removing the Appendix A from the paper, but 1) retaining Appendix-Figure 5 as a new figure, and provide the alignment used for constructing the phylogeny tree as a supplementary data; 2) retaining Table Appendix A and making it a new Supplementary table.

We appreciate the feedback and believe that such changes improve the manuscript. We have removed Appendix A, leaving Figure 5 from that section (with the addition of the requested alignment information) in the main text, and the table also present in that section, as a new Supplemental Table.

2) Figure 2A is not necessary. Fig. 2B, it is better to use kb, and please enlarge the font so it is readable.

We have removed Figure 2A and improved Figure 2B, which in the new version is Figure 2.

3) Figure S2 is not necessary.

As suggested by the reviewer, we have removed Figure S2.

4) Fig. 4 legend: it is difficult to understand the following sentence “Between parentheses (in black letters) it is indicated how many of the shared protein genes form groups of orthologs with sequences present in all viruses considered”, and I do not understand the related numbers in Figure 4.

We have changed the text of the legend to make it easier to understand. We agree that the original text is confusing. We hope that the new proposal will facilitate understanding.

5) Figure 5 and related description are not necessary.

As requested by the reviewer, we have decided to remove Figure 5 from the main text and make it a supplementary figure.

6) Table S2 is not necessary; it can be easily included in Table S4.

Tables S1 and S2 have been merged in the new version of the manuscript.

7) Table S4. It is better to put SFHV together instead of separate them in the table. I did not check the data in detail, but find that P33 homologues should be present in most of the invertebrate viruses, but it was N in the table. Please check the data carefully.

We have rearranged the columns of the table in accordance with what was opportunely observed by the reviewer. The results of Table S4 are based on the annotated proteins of the studied genomes detected by our working algorithm. The absence of detected orthologs does not necessarily mean that these viruses lack the corresponding proteins (they could be false negatives). However, the presence of a core gene in one of the other insect viruses does indicate that this core gene has a conserved ortholog in that virus. Regarding P33 specifically, we performed a Blastp search using the baculoviral P33 as a database and all annotated proteins from the other insect viruses as queries. Although we obtained hits, they had very poor significance values (E-value=0.033), which may be the reason why our algorithm did not detect them. Lowering the thresholds to select positive results could be a potential improvement point for the algorithm.

8) Result 3.3, I found the entire section unnecessary, as baculovirus phylogeny have been extensively studied, and this study did not add any new information.

We agree with the reviewer that the value of phylogeny with the 38 core genes only lies in having used a greater number of genomes. In this sense, we have eliminated Figure 6A and have proposed a new Figure 6 focusing on the findings in MOBP.

9) Abstract, “more than 250 genome”, change to “nearly 300 genomes”

We appreciate the suggestion and have incorporated it.

10) Introduction: Our knowledge of the two phenotypes of baculovirus, ie. BV and ODV, are largely generated from lepidopteran baculoviruses (alphabaculovirus and betabaculovirus). Currently it is generally believed that gammabaculoviruses do not contain a BV life cycle, and it remains unknown if deltabaculovirus contains BV. Please modify the related description in the manuscript, such as line 33-36.

We appreciate and agree with the observation. Perhaps our way of expressing it has not been adequate, so we have changed that part of the text in accordance with the knowledge available up to now.

11) Introduction: Line 39-40. “OBs may contain 1 or more ODVs, and these in turn may contain 1 or more nucleocapsids depending on the baculoviral species”. This is not a correct description, as GV normally only contain 1 ODV.

We regret that our description has not been adequate, as we believe that it contained what was observed by the reviewer. In keeping with the fact that the phrase can be misleading, we have made changes in the new version of the manuscript.

12) Introduction: line 74-75:” the number of homologous genes (known as core genes)”, this definition is incorrect. Core genes are the genes that are conserved in all the baculovirus genomes. While homologous genes are the ones that can find homologues in other baculovirus genomes. Please correct.

We appreciate and agree with the comment. We have made the changes to the text.

13) Line 238, change “wide limit” to “wide range”

We have made the correction.

14) Line 241, delete “thus”

We have made the correction.

15) Line 255, “the other ones have greater in AT in similar values”, please revise the sentence to make it easier to understand

We have made a change to the text in that section.

16) Line 276-277 please remove “probably derived from the coevolution of these viruses with arthropods and their expansion throughout the entire planet”, as it is more suitable in the discussion section than in the result section.

As suggested, the phrase was removed.

17) Line 356, what is “delta- y gammabaculoviruses”?

Unfortunately, it was a mistake to incorporate a word in Spanish. This has been fixed.

18) Discussion, the current discussion is quite long and by removing the phylogeny results, the discussion needs to be revised.

We have revised and modified part of the discussion.

19)   The English writing needs to be improved.

We have carried out a thorough revision of the English throughout the entire manuscript.

Round 2

Reviewer 1 Report

The authors made a substantial revision of their initial submission, trying to address some of the reviewers´ concerns. However, the manuscript still contains major flaws and more than questionable analyses and conclusions as I will address in the following:

One of my concerns was that the content of novelty is limited because similar results were obtained with a smaller data set. The authors argue in their rebuttal that the increase of sequenced genomes would warrant a re-analysis. In the discussion (line 727 pp), they even claim an “exponential increase of the availability of genome sequences”. To my understanding the analysis is flawed by this “exponential increase” as the increase in recent years is mainly based on re-sequencing of a large number of isolates from a very few baculovirues species. About more than half of their dataset derive from less than 10 baculovirus species (Table S1), i.e. BmNPV, AgMNPV, ChchNPV, HearNPV, LdMNPV, MacoNPV, SeMNPV, CpGV, PhopGV and PlxyGV! Therefore, the calculated genome lengths and GC contents (Fig. 2) and gene contents (Fig. S5) do not contain new information but are even biased by the equal weight of baculovirus isolates and baculovirus species in the analyses. Since intraspecies variability of virus isolates is much lower than interspecies variability in terms of GC content, gene content, genome length, the data set is not balanced but the weight of these 10 species alone make about 50% of the analyses. For a proper analyses each species within a genus should enter the calculation with the same weight and it should not depend on the chance how many isolates of given baculovirus were sequenced. Although the authors note in line 76pp that there might be an unequal unbalanced distribution of the known baculoviruses because hosts of alpha- and betabaculovirus are usually important major agricultural pests worldwide, they bias their analyses additionally by giving the 10 economically most important baculoviruses about the same weight as all other together.

I am most concerned about the approach taken by the authors of putting all 297 genomes together into statistical analyses also only 107 baculoviruses were significant different (different species ) (line 698). Thus two thirds of the analysed data set is more or less redundant, at least for the statistical analyses performed.

 My second major concern was the claimed homology of Cun085 and the Polh/Gran of Alpha-, Beta- and Gammabaculoviruses, which were identified as MOBP. The presentation was not convincing in the first submission and is not convincing in the revised version. To support their claim, two alignments are now included (Fig. 4 and Fig. 6).

To my understanding an alignment is a hypothesis on the homology of a nucleotide or amino acid position in a sequence that is based on identity or biochemical similarity of a residue. If so, how can the alignment of Cun085 with Polh/Gran can be completely different in Fig. 4 and Fig. 6 and again differ from the former analysis in the original submission? The identified structural motifs in Fig. 4 cannot be identified again in Fig. 6 (just compare NLS). Both Figures 4 ad 6 in the revision as well as the original submission contradict each other and exemplify the weakness and arbitrariness of the aligned amino acid residues.

By comparing with some other baculovirus core genes (desmoplakin, DNApol, helicase, p6.9) which are highly variable (Fig. 5) it is argued that Cun085 and the Polh/Gran may also exhibit a hidden homology. The chosen examples, however, show a continuum of identity and similarity among different isolates from four genera, whereas in case of Cun085, such continuum does not exist. All Polh/Gran from Alpha-, Beta-, and Gammabaculoviruses show sequence similarity of >75%, whereas that to Cun085 does not exceed 15% (Fig. 5B). Sequence identity between Cun085 and appears to be around 5% (Fig. 5A) which could be expected from any random alignment with any sequence. The proposed homology of Cun085 and Polh/Gran is not justified by the presented analyses.

Further Comments:

Line 34 and line717: reference to Baltimore classification is redundant. Baltimore classification is not only the "type of genomic nucleic acid” (line 717) but also refers to the process how it is expressed. Otherwise, there would be no class 6 (+RNA genome) and class 7 (gapped ds DNA genome). Instead of referring to Baltimore classification the authors should refer to new classification including the class of Naldaviricetes and the order Lefavirales.

Line 41: “two virion variants in some baculoviruses”. Be more specific, which Baculoviruses do not produce BVs?

Line 45 and elsewhere: numbers between 1 and 12 should be written out.

Line 55: “F protein”

Line 338. “Culex nigripalpus NPV, has a genome of 108,252 bp, which is close to the average length for betabaculoviruses.” Has this statement any biological meaning?

Line 343. How were “prototypes” identified or defined?

Lines 346-360: What is the biological meaning or novelty of these findings?

Line 355: Why should that be a unique property of the Gammabaculovirus genus, when the genus harbors only one species? If there are more gammabaculoviruses, there might be also variation in the GC content.

 Line 358-359: The baculovirus genome length or GC content was never considered as a classification criterion, therefore the thereof claimed necessity of using gene content for classification does not really exist but is common practice since the seminal publication of Herniou et al. 2001.

Line 381: Delete sentence.

Line 389: “much more convergent in their common ancestry?“

Convergence means independent development of analogous characters. Alpha- and Betabaculoviruses did not diverge as much as from Gamma- and Deltabaculoviruses.

Line 408 and elsewhere: „numbers“ instead of „amounts“

Line 425: It was not presented that Ortholog Bioinformatic Pipeline succeeded in identification of CUN085 as Polh/Gran homolog. Fig. S1 shows the pipline´s result only for the 38 core genes.

Figure 4: I am wondering of the structural motifs where identified on the bases of the four prototypes (AGI, AGII, B, G) or if they would also be recognized for Cun085 sequence alone? I cannot recoginze conservation of any of these regions. Cristallography analyses of NPVs and GVs OB protein revealed conserved functional domains and amino acid residues of Wiseana NPV and CpGV Polh/Gran. It might be interesting to check if those could be recognized in Cun085.

Line 455: “Even the region containing the nuclear localization signal (NLS) is shown to be conserved.” It is not surprising for a structural baculovirus protein to have a NLS, because the virion/OB assembly is in the nucleus. It might be the only shared domain in the complete protein.

Figure 6 A. shows a phylogram not a cladogram. It is not clear which genes from other Lefavirales were used to establish the tree, neither the method for tree construction is indicated. 6B. The alignment of amino acid positions within baculoviruses differs completely from Fig. 4!

Line 626: Because of the weight of the 38 core genes in the analyses, it would be more than surprising if addition of Cun085 and Polh/Gran could make any difference except for the length of the CuniNPV branch in Figure 7.

Figure 7: Bootstrap colours in the center of the figure are not explained. Numerous of such trees based on baculovirus core gene analyses have been published. This tree just contains the re-sequenced isolates (see above) which all do contribute to the large picture of baculovirus phylogeny. I cannot recognize any novelty and value in this tree.

Figure 8: For me this analysis again does not make real sense because it does not contribute to species identification but is again blurred by mixing re-sequenced isolates and species. The species within different genera are so distant that the K2P distance of course has to exceed the value of 1. The whole intention of this analysis is not clear.

Line 684: The median again depends on the number of re-sequences isolates within a genus and has no value.

Line 684-691 does not contain any new information

Line 692-698: Argumentation is weird. It is not that isolates were classified into species but that different isolates form certain species were sequenced. So, all these “results” are not really significant except that other researchers where correct in allocating their sequenced isolates to a given species.

Line 699: I am not aware of any baculovirus isolates with low K2P values and following the established species distance criteria, which had been separated to two species based on gene content. This is only necessary for those which do not meet the species criteria.

Line 708pp: Discussion could be significantly shortened. Reference to LUCA and LECA not meaningful at all, esp. as viruses do not have a common ancestor.

Line 717: reference to Baltimore classification redundant to introduction and wrongly defined. See above.

Line 725-727: Missing reference to Naldaviricetes and Lefavirales. “exponential increase” only because of re-sequencing isolates. The information gained from that is not on the level of interspecies relationship but on intraspecific variation. Therefore, these re-sequenced isolates hardly contribute to aim of the publication.

Line743-745: As outlined above, I doubt that this can be concluded from presented data.

Line 746-748: How could it “compromise“ phylogenetic analyses, as distance is mainly determined by the other 38 core genes and including Cun085 would make distance even larger.

Line 750: this statement is only true for OB producing lefaviral viruses.

Line 768 to 770: Statement is contradicted by Fig. 4 and Fig. 6

Author Response

Dear Reviewer #1,

In response to your comments and concerns, we offer the following answers (in blue).

The authors made a substantial revision of their initial submission, trying to address some of the reviewers´ concerns.

We appreciate the thorough review performed on our manuscript. The observations, doubts and opinions expressed help us to improve our work. Below we will respond to each of these concerns.

However, the manuscript still contains major flaws and more than questionable analyses and conclusions as I will address in the following:

One of my concerns was that the content of novelty is limited because similar results were obtained with a smaller data set. The authors argue in their rebuttal that the increase of sequenced genomes would warrant a re-analysis. In the discussion (line 727 pp), they even claim an “exponential increase of the availability of genome sequences”. To my understanding the analysis is flawed by this “exponential increase” as the increase in recent years is mainly based on re-sequencing of a large number of isolates from a very few baculovirues species. About more than half of their dataset derive from less than 10 baculovirus species (Table S1), i.e. BmNPV, AgMNPV, ChchNPV, HearNPV, LdMNPV, MacoNPV, SeMNPV, CpGV, PhopGV and PlxyGV! Therefore, the calculated genome lengths and GC contents (Fig. 2) and gene contents (Fig. S5) do not contain new information but are even biased by the equal weight of baculovirus isolates and baculovirus species in the analyses. Since intraspecies variability of virus isolates is much lower than interspecies variability in terms of GC content, gene content, genome length, the data set is not balanced but the weight of these 10 species alone make about 50% of the analyses. For a proper analyses each species within a genus should enter the calculation with the same weight and it should not depend on the chance how many isolates of given baculovirus were sequenced. Although the authors note in line 76pp that there might be an unequal unbalanced distribution of the known baculoviruses because hosts of alpha- and betabaculovirus are usually important major agricultural pests worldwide, they bias their analyses additionally by giving the 10 economically most important baculoviruses about the same weight as all other together.

I am most concerned about the approach taken by the authors of putting all 297 genomes together into statistical analyses also only 107 baculoviruses were significant different (different species ) (line 698). Thus two thirds of the analysed data set is more or less redundant, at least for the statistical analyses performed.

In recent years, new baculoviral genomes have been added to the GenBank repository, some of which have accompanying papers while others do not. Among these new additions, some are isolates of previously described species while others represent new species (at least according to the K2P-based distance criteria). The previous study that established comparison ranges based on K2P worked with 172 genomes (Wennmann JT, Keilwagen J, Jehle JA. Baculovirus Kimura two-parameter species demarcation criterion is confirmed by the distances of 38 core gene nucleotide sequences. J. Gen. Virol. 2018, 99, 1307-1320), while in our study we used 297. Although some of the new additions belong to genotypes of described species, others would form new species, and we propose that these new additions warrant a genomic and genetic reanalysis of baculoviruses. Our study determines 107 species (Alphabaculovirus: 76; Betabaculovirus: 26; Gammabaculovirus: 3; Deltabaculovirus: 1) vs 69 species presented in the mentioned paper (Alphabaculovirus: 40; Betabaculovirus: 25; Gammabaculovirus: 3; Deltabaculovirus: 1). We acknowledge that incorporating multiple isolates for the same species could introduce bias in comparative studies. However, we repeated the studies by only considering one genome per species (we identified 107 baculoviral species) and observed that the trends remained unchanged. This additional analysis is presented in a new supplementary figure. It is also worth clarifying, as we specified in the Methodology section, that we established criteria to select genomes for our study that had differences in their sequences. Therefore, even if they are members of the same species, they may also be genotypic variants.

My second major concern was the claimed homology of Cun085 and the Polh/Gran of Alpha-, Beta- and Gammabaculoviruses, which were identified as MOBP. The presentation was not convincing in the first submission and is not convincing in the revised version. To support their claim, two alignments are now included (Fig. 4 and Fig. 6).

To my understanding an alignment is a hypothesis on the homology of a nucleotide or amino acid position in a sequence that is based on identity or biochemical similarity of a residue. If so, how can the alignment of Cun085 with Polh/Gran can be completely different in Fig. 4 and Fig. 6 and again differ from the former analysis in the original submission? The identified structural motifs in Fig. 4 cannot be identified again in Fig. 6 (just compare NLS). Both Figures 4 ad 6 in the revision as well as the original submission contradict each other and exemplify the weakness and arbitrariness of the aligned amino acid residues.

The different alignments shown in Figures 4 and 6 were generated to answer different types of questions, so they were performed on different sets of sequences, which is why they are not the same. In the case of Figure 4, the alignment was made with the MOBPs of the 297 baculoviruses, including the CuniNPV MOBP. The alignment was then manually curated taking into account the secondary structure prediction information (alpha helices and beta sheets) of the MOBP proteins of the four prototype genomes and CuniNPV, determining those regions where there was structural conservation beyond sequential. To generate a more pleasing image, only the MOBPs of the baculoviral prototypes are shown in Figure 4. Therefore, the alignment in Figure 4 is guided by the conservation of secondary structure (both in CuniNPV and in the other prototypes), and the objective of the alignment was to show that, to our knowledge, despite the low sequence conservation, there seems to be significant conservation in secondary structure and function, so they could be considered homologs.

On the other hand, the alignment in Figure 6 is made for purely phylogenetic purposes. The question to be answered was what the evolutionary relationship was between the MOBPs of baculoviruses and those of other arthropod viruses, in addition to evaluating whether the CuniNPV MOBP was closer to the other baculoviruses or closer to viruses from other families. Therefore, in this case, the sequences used are different, and the alignment was made from a multiple Fasta with nine MOBP sequences (five from baculoviruses and four from other arthropod viruses). Manual curation was not done based on conservation of secondary structure. Finally, as the reviewer rightly indicates, the new alignment (Figure 4) is different from the previous one because this alignment now has manual curation based on structural conservation, while in the alignment of the first version, this analysis was not done. This analysis was added based on suggestions from the reviewers, and in the new version we also added the information of tertiary structure.

By comparing with some other baculovirus core genes (desmoplakin, DNApol, helicase, p6.9) which are highly variable (Fig. 5) it is argued that Cun085 and the Polh/Gran may also exhibit a hidden homology. The chosen examples, however, show a continuum of identity and similarity among different isolates from four genera, whereas in case of Cun085, such continuum does not exist. All Polh/Gran from Alpha-, Beta-, and Gammabaculoviruses show sequence similarity of >75%, whereas that to Cun085 does not exceed 15% (Fig. 5B). Sequence identity between Cun085 and appears to be around 5% (Fig. 5A) which could be expected from any random alignment with any sequence. The proposed homology of Cun085 and Polh/Gran is not justified by the presented analyses.

In this case, it is true that the identity and similarity values of CUN085 are low. However, there is functional similarity for MOBP that has not been determined in other CuniNPV genes that are considered within the core gene group. Similar cases occur with MOBP from other invertebrate viruses. In nudiviruses there is also a MOBP that, despite not having apparent sequential similarity with the other ones, does have functional similarity [Chaivisuthangkura P, Tawilert C, Tejangkura T, Rukpratanporn S, Longyant S, Sithigorngul W, Sithigorngul P. (2008). Molecular isolation and characterization of a novel occlusion body protein gene from Penaeus monodon nucleopolyhedrovirus. Virology, 381(2), 261–267; Yang YT, Lee DY, Wang Y, Hu JM, Li WH, Leu JH, Chang GD, Ke HM, Kang ST, Lin S S, Kou GH, Lo CF (2014). The genome and occlusion bodies of marine Penaeus monodon nudivirus (PmNV, also known as MBV and PemoNPV. BMC genomics, 15(1), 628]. Just as the syntenic organization of genes is an accepted property for considering common ancestry, the function of the encoded protein within a lineage of related viral nucleic acids (as in the case of Naldaviricetes) should be given even more consideration. This is particularly relevant for MOBP, where partially conserved sequence regions can be detected, as we demonstrated in our study.

Further Comments:

Line 34 and line717: reference to Baltimore classification is redundant. Baltimore classification is not only the "type of genomic nucleic acid” (line 717) but also refers to the process how it is expressed. Otherwise, there would be no class 6 (+RNA genome) and class 7 (gapped ds DNA genome). Instead of referring to Baltimore classification the authors should refer to new classification including the class of Naldaviricetes and the order Lefavirales.

We appreciate the observation. This has been added in the text.

Line 41: “two virion variants in some baculoviruses”. Be more specific, which Baculoviruses do not produce BVs?

We have added more detail about it.

Line 45 and elsewhere: numbers between 1 and 12 should be written out.

We made the requested change to the text.

Line 55: “F protein”

We have introduced it in the text.

Line 338. “Culex nigripalpus NPV, has a genome of 108,252 bp, which is close to the average length for betabaculoviruses.” Has this statement any biological meaning?

This comparison aims to only compare one genomic descriptor (the length of viral nucleic acid nucleotides) between genera. This does not necessarily imply a biological link, but it also does not rule out the possibility of one. In many viruses, genomic dimensions are linked to virion dimensions, and relationships exist between assembled protein complexes and nucleic acid length. However, in this particular case, none of this can be determined. Nevertheless, the identification of a shared characteristic, although it may be a chance finding, could serve as the starting point to describe the possible underlying mechanisms responsible for this property. Furthermore, all these properties may be of interest to engineer these viruses for biotechnological applications.

Line 343. How were “prototypes” identified or defined?

In this section of the manuscript, the bibliographic citation is introduced where the type species for each genus are proposed (Jehle, J.A.; Blissard, G.W.; Bonning, B.C.; Cory, J.S.; Herniou, E.A.; Rohrmann, G.F.: Theilmann, D.A.; Thiem, S.M.; Vlak, J.M. On the classification and nomenclature of baculoviruses: a proposal for revision. Arch. Virol. 2006, 151, 1257-1266).

Lines 346-360: What is the biological meaning or novelty of these findings?

The growth of the genomic sequence database necessitates regular reanalysis to update the limits and averages of the quantitative properties being studied. The novelty of this study lies in its reliance on up-to-date data.

Line 355: Why should that be a unique property of the Gammabaculovirus genus, when the genus harbors only one species? If there are more gammabaculoviruses, there might be also variation in the GC content.

Based on the criterion of genetic distance between conserved genes, gammabaculoviruses are currently classified into three species, which share a similar percentage of GC content. It is important to note that this classification may change as new species are discovered and described. However, until then, the %GC content within this genus can be considered a shared property among its members.

 Line 358-359: The baculovirus genome length or GC content was never considered as a classification criterion, therefore the thereof claimed necessity of using gene content for classification does not really exist but is common practice since the seminal publication of Herniou et al. 2001.

We agree with the comment made by the reviewer, and our intention is to introduce that concept into that sentence. General properties of viral nucleic acids, such as length, GC content, repeats, structure, etc., can become taxonomic descriptors if they are linked to some biological property underlying the infection cycle. Clearly, this is not the case for baculoviruses (or at least requires more in-depth study to establish a consensus), so the information content at the level of encoded proteins becomes a variable that can be used to understand their evolution.

Line 381: Delete sentence.

The sentence was deleted.

Line 389: “much more convergent in their common ancestry?“

Convergence means independent development of analogous characters. Alpha- and Betabaculoviruses did not diverge as much as from Gamma- and Deltabaculoviruses.

We understand that this was not the best way to explain the result, and we appreciate the feedback. We have made changes to the text accordingly.

Line 408 and elsewhere: „numbers“ instead of „amounts“

The change was incorporated.

Line 425: It was not presented that Ortholog Bioinformatic Pipeline succeeded in identification of CUN085 as Polh/Gran homolog. Fig. S1 shows the pipline´s result only for the 38 core genes.

Figure S1 shows only the results related to the calibrator group for the pipeline (the 38 accepted core genes). However, the pipeline also enabled us to identify another set of genes where only the homologue in deltabaculovirus remained to be proposed, such as mobp and lef11. In these cases, synteny analysis was performed (equivalent to that conducted for p6.9 and desmoplakin), and the literature was searched for reports that assigned a biological function to the CuniNPV proteins. Accordingly, CUN085 was included for subsequent analysis.

Figure 4: I am wondering of the structural motifs where identified on the bases of the four prototypes (AGI, AGII, B, G) or if they would also be recognized for Cun085 sequence alone? I cannot recoginze conservation of any of these regions. 

Figure 4 displays the secondary structures of CuniNPV MOBP in comparison with the secondary structures of the other 4 baculoviral proteins that were obtained from the baculoviruses suggested as prototypes. The darker color filling in the same figure indicates those secondary structures that occur in all proteins. In particular, Figure 4 highlights the following conserved structures: 8 alpha-helices and 7 beta-sheets.

Cristallography analyses of NPVs and GVs OB protein revealed conserved functional domains and amino acid residues of Wiseana NPV and CpGV Polh/Gran. It might be interesting to check if those could be recognized in Cun085.

Taking into consideration the reviewer's suggestion, we have marked in the alignment of Figure 4 (with yellow circles) those residues previously determined to be important in structural analyses of MOBPs, and which are conserved in our alignment [Coulibaly F, Chiu E, Gutmann S, Rajendran C, Haebel PW, Ikeda K, Mori H, Ward VK, Schulze-Briese C, Metcalf P. The atomic structure of baculovirus polyhedra reveals the independent emergence of infectious crystals in DNA and RNA viruses. Proc Natl Acad Sci U S A. 2009 Dec 29;106(52):22205-10; Ji X, Sutton G, Evans G, Axford D, Owen R, Stuart DI. How baculovirus polyhedra fit square pegs into round holes to robustly package viruses. EMBO J. 2010 Jan 20;29(2):505-14]. Likewise, it was possible to observe that the Cysteine involved in the formation of the disulfide bridge (now marked in bold in Figure 4) and the amino acids involved in the intermolecular salt bridges, which hold together two MOBP trimers, are conserved. In the future, it will be important to study the tertiary structure of CUN085 in a comparative manner to provide useful information regarding their common ancestry with the other MOBPs.

Line 455: “Even the region containing the nuclear localization signal (NLS) is shown to be conserved.” It is not surprising for a structural baculovirus protein to have a NLS, because the virion/OB assembly is in the nucleus. It might be the only shared domain in the complete protein.

It is true that it is not surprising, but it is one of the characteristics that are shared. However, as made clear above, they are not the only shared functional residues.

Figure 6 A. shows a phylogram not a cladogram.

Thank you for the observation. We apologize for the error, as what is shown is a phylogram. We have made the necessary correction in the text.

It is not clear which genes from other Lefavirales were used to establish the tree, neither the method for tree construction is indicated. 6B.

The MOBP proteins to make the alignment were selected based on the paper by Yang et al [Yang YT, Lee DY, Wang Y, Hu JM, Li WH, Leu JH, Chang GD, Ke, HM, Kang ST, Lin SS, Kou GH, Lo CF (2014). The genome and occlusion bodies of marine Penaeus monodon nudivirus (PmNV, also known as MBV and PemoNPV) suggest that it should be assigned to a new nudivirus genus that is distinct from the terrestrial nudiviruses. BMC genomics, 15(1), 628.]. Appropriate GenBank searches (Blastp and tBlastn) were then performed to confirm that they were MOBP, and the original papers for each considered species were reviewed. Unfortunately, the information on the accession numbers of the proteins used was omitted due to an error. In this new version, they have been added to Figure 6.

The alignment of amino acid positions within baculoviruses differs completely from Fig. 4!

The different alignments shown in Figures 4 and 6 were generated to answer different types of questions, so they were performed on different sets of sequences. Therefore, they are not the same.

Line 626: Because of the weight of the 38 core genes in the analyses, it would be more than surprising if addition of Cun085 and Polh/Gran could make any difference except for the length of the CuniNPV branch in Figure 7.

We agree with the reviewer's observation. However, phylogenetic inferences based on individual proteins do not always propose the same evolutionary hypothesis as the use of a concatenation of shared proteins. Furthermore, if mobp is a gene derived from a common ancestor in Baculoviridae, it should be included in the evolutionary study, and the variations it has accumulated should be considered in the resulting phylogenetic inferences. This information must be retained if new core genes are added in the future.

Figure 7: Bootstrap colours in the center of the figure are not explained. Numerous of such trees based on baculovirus core gene analyses have been published. This tree just contains the re-sequenced isolates (see above) which all do contribute to the large picture of baculovirus phylogeny. I cannot recognize any novelty and value in this tree.

We apologize for not including bootstrap information in the figure legend. This has now been fixed. Moreover, this inference is not only the most up-to-date among all reported so far, including all the isolates sequenced up to the date indicated in the manuscript, but it is also the first to be carried out with 39 core proteins. In our opinion, any modification in the number of shared genes in Baculoviridae must be accompanied by a phylogenetic study. Furthermore, although a large number of the sequences correspond to isolates of the same species, new species are also included, as mentioned above.

Figure 8: For me this analysis again does not make real sense because it does not contribute to species identification but is again blurred by mixing re-sequenced isolates and species. The species within different genera are so distant that the K2P distance of course has to exceed the value of 1. The whole intention of this analysis is not clear.

The purpose of this analysis is to demonstrate the ranges of genetic distances between all the sequenced baculoviral isolates. In our opinion, this information is valuable in verifying the classification of currently accepted taxa and can potentially identify new subtaxa in the future based on narrower distance ranges. Additionally, the existence of multiple isolates for the same species is represented in the graph as horizontal dotted lines, indicating which genera have this type of information available and which do not. This graph complements the evolutionary information proposed through phylogenetic inferences, as it is based on different analyses.

Line 684: The median again depends on the number of re-sequences isolates within a genus and has no value.

We agree that the statistics derived from a dataset depend on its composition. To provide useful information, we added a new supplementary figure that reanalyzes the parameters compared to the database filtered by the number of species after the K2P study was performed. As shown in the new figure, the trends reported in the initial analyses (without eliminating isolates of the same species) are preserved.

Line 684-691 does not contain any new information

We respect the comment of the reviewer, but we consider, as we mentioned before, that this analysis complements the phylogenetic study and offers useful information.

Line 692-698: Argumentation is weird. It is not that isolates were classified into species but that different isolates form certain species were sequenced. So, all these “results” are not really significant except that other researchers where correct in allocating their sequenced isolates to a given species.

The paragraph in question identifies the number of baculovirus species based on widely accepted criteria and the diversity of available genomes. Some of the isolates used in our analysis have been identified in previous studies as belonging to a particular species, while others have not. The aim of this analysis is to bring together the diversity of species within Baculoviridae into a single analysis, which is useful for those studying viral evolution. For instance, it could facilitate the selection of a representative isolate per species for future genomic studies.

Line 699: I am not aware of any baculovirus isolates with low K2P values and following the established species distance criteria, which had been separated to two species based on gene content. This is only necessary for those which do not meet the species criteria.

This sentence is in line with the currently accepted principle for classifying baculoviruses into species, which is based on the genetic distance from the ancestrally shared genome. However, there could be isolates with very similar genomes but whose changes are sufficient to, for example, alter the morphology of the virion or its biological effects (host range, LD50, LT90, etc.). Our research group has characterized an NPV isolated from Spodoptera ornithogalli, which presents K2P values close to those of the NPVs isolated from Spodoptera frugiperda but shows significant morphological and biological changes (Barrera et al. Natural Coinfection between Novel Species of Baculoviruses in Spodoptera ornithogalli Larvae. Viruses. 2021 Dec 15;13(12):2520. doi: 10.3390/v13122520). In addition, our research group also reported a NPV of Spodoptera frugiperda with a differential genetic content to the rest described for the same species (Barrera et al. Evidence of recent interspecies horizontal gene transfer regarding nucleopolyhedrovirus infection of Spodoptera frugiperda. BMC Genomics. 2015 Nov 25;16:1008. doi: 10.1186/s12864-015-2218-5). For this reason, and since the classification of viruses into species is a human cultural construct made up of multiple components, all of them could be considered for a better description of the viral diversity present in nature.

Line 708pp: Discussion could be significantly shortened. Reference to LUCA and LECA not meaningful at all, esp. as viruses do not have a common ancestor.

We appreciate the reviewer's feedback on the length and content of the discussion, but we believe it contains the appropriate reflections based on the analyses conducted. Additionally, we agree that the diversity of viruses on Earth does not have a common ancestor as organisms do. Viral nucleic acids, along with other nucleic acids such as transposons and plasmids, can be considered part of the mobilome, with multiple ancestral connections to each other. It is likely that many DNA viruses share common ancestry, such as dsDNA viruses that infect arthropods. In this sense, common ancestral inferences could be proposed for these lineages. This is a topic of interest to our research group, and thus, we drew an analogy with LUCA and LECA. We believe this mention is appropriate since it is part of our working hypotheses, which we will continue to explore.

Line 717: reference to Baltimore classification redundant to introduction and wrongly defined. See above.

This was modified.

Line 725-727: Missing reference to Naldaviricetes and Lefavirales. “exponential increase” only because of re-sequencing isolates. The information gained from that is not on the level of interspecies relationship but on intraspecific variation. Therefore, these re-sequenced isolates hardly contribute to aim of the publication.

The mention of the exponential increase in sequences in that section of the manuscript is not linked only to baculoviruses. Instead, it refers to the increase in global genomic information, including viruses and organisms. There is a need to build new computational algorithms to enable the analysis of this vast amount of data. In the case of baculoviruses, this also includes an increase in knowledge of variants within the same species. However, the consideration of these sequences is valuable because even within the same baculoviral species, there may not only be allelic differences but also differences in gene content, as mentioned before. Furthermore, our work systematizes all the information on protein-coding genes, offering the community identification of each coding sequence of almost 300 genomes, and their linkage by hypothetical common ancestry between (orthology) and within genomes (paralogy). We would like to apologize for the omission of the recent classification accepted by ICTV regarding the Class Naldaviricetes and the new order Lefavirales. We have included this information in the new version of the manuscript.

Line743-745: As outlined above, I doubt that this can be concluded from presented data.

We acknowledge the reviewer's opinion, but we believe that the evidence we have presented is sufficient to propose mobp as a potential new core gene. Only a few proteins in CuniNPV have been functionally characterized, and CUN085 is one of them. The sequence homology observed between CUN085 and its functional protein homologues (polyhedrin/granulin) suggests that it is not the result of chance and is unlikely to be a product of convergent evolution.

Line 746-748: How could it “compromise“ phylogenetic analyses, as distance is mainly determined by the other 38 core genes and including Cun085 would make distance even larger.

As we have mentioned before, the incorporation of any sequence in a phylogenetic study can alter the distribution and relationships between the entities being considered. We agree with the idea that the weight of a sequence in a concatemer with many others may not provide enough value to offer a different inference. However, the addition of mobp was necessary given our proposal as a new core gene.

Line 750: this statement is only true for OB producing lefaviral viruses.

In light of this comment, CuniNPV belongs to the order Lefavirales. Therefore, we believe that CUN085 is not a product of convergent evolution and propose that its incorporation into the same orthology group as the other OB-forming proteins is justified based on sequence similarity and common function.

Line 768 to 770: Statement is contradicted by Fig. 4 and Fig. 6

We do not consider our statements to be contradictory. We summarize the observations made about MOBP as it was being described and demonstrate that despite the divergence observed in CUN085 and other representatives of Naldaviricetes, traces of their possible common ancestry are still preserved.

Reviewer 3 Report

The revision has significantly improved. I still have difficulty to understand Figure 3. I suggest: 1) in the text, use an example to explain what the numbers represent for; 2) in the figure, choose another color to distinguish the bold black letter and the black letter. 

Author Response

Dear Reviewer #1,

In response to your comments and concerns, we offer the following answers (in blue).

The revision has significantly improved. I still have difficulty to understand Figure 3. I suggest: 1) in the text, use an example to explain what the numbers represent for; 2) in the figure, choose another color to distinguish the bold black letter and the black letter. 

We appreciate the reviewer's consideration of the changes made. Regarding Figure 3 and its legend, we made changes in this new version to facilitate its understanding.
